# ORP5 and ORP8 bind phosphatidylinositol-4, 5-biphosphate (PtdIns(4,5)$P_2$) and regulate its level at the plasma membrane

Rajesh Ghai [1,5], Ximing Du[1], Huan Wang[2], Jiangqing Dong[2], Charles Ferguson[3,4], Andrew J. Brown[1], Robert G. Parton [3,4], Jia-Wei Wu[2] & Hongyuan Yang [1]

ORP5 and ORP8, members of the oxysterol-binding protein (OSBP)-related proteins (ORP) family, are endoplasmic reticulum membrane proteins implicated in lipid trafficking. ORP5 and ORP8 are reported to localize to endoplasmic reticulum–plasma membrane junctions via binding to phosphatidylinositol-4-phosphate (PtdIns(4)$P$), and act as a PtdIns(4)$P$/phosphatidylserine counter exchanger between the endoplasmic reticulum and plasma membrane. Here we provide evidence that the pleckstrin homology domain of ORP5/8 via PtdIns(4,5)$P_2$, and not PtdIns(4)$P$ binding mediates the recruitment of ORP5/8 to endoplasmic reticulum–plasma membrane contact sites. The OSBP-related domain of ORP8 can extract and transport multiple phosphoinositides in vitro, and knocking down both ORP5 and ORP8 in cells increases the plasma membrane level of PtdIns(4,5)$P_2$ with little effect on PtdIns(4)$P$. Overall, our data show, for the first time, that phosphoinositides other than PtdIns(4)$P$ can also serve as co-exchangers for the transport of cargo lipids by ORPs.

[1] School of Biotechnology and Biomolecular Sciences, The University of New South Wales, Sydney, NSW 2052, Australia. [2] Beijing Advanced Innovation Center for Structural Biology, MOE Key Laboratory for Protein Science, Tsinghua-Peking Center for Life Sciences, School of Life Sciences, Tsinghua University, Beijing 100084, China. [3] Centre for Microscopy and Microanalysis, The University of Queensland, St. Lucia, QLD 4072, Australia. [4] Institute for Molecular Bioscience, The University of Queensland, St. Lucia, QLD 4072, Australia. [5] Present address: Institute for Molecular Bioscience, The University of Queensland, St. Lucia, QLD 4072, Australia. Rajesh Ghai, Ximing Du and Huan Wang contributed equally to this work. Correspondence and requests for materials should be addressed to J.-W.W. (email: jiaweiwu@mail.tsinghua.edu.cn) or to H.Y. (email: h.rob.yang@unsw.edu.au)

Cellular compartmentalization into membranous organelles requires precise spatio-temporal distribution of certain lipids that serve as organelle identity signatures[1]. The intracellular trafficking of lipids is therefore central to normal cellular homeostasis. Recent studies show that specific non-vesicular lipid transfer pathways play crucial roles in the maintenance of membrane lipid composition[2, 3]. In particular, dynamic endoplasmic reticulum (ER) membrane tubules spread throughout the cell to form close physical contacts with other organelles[4]. These membrane contact sites (MCSs) are separated by gaps in the range of 23–25 nm[5] and are highly enriched in lipid transfer proteins (LTPs), which are known mediators of non-vesicular lipid transport[6].

The oxysterol-binding protein (OSBP) and its related proteins (ORP, for OSBP-related protein) have emerged as central regulators of sterol/lipid transport at the junctions formed by ER with other organelles[7–10]. OSBP and its homologs are conserved from yeast (Osh family) to mammals (ORP family)[11, 12]. OSBP was recently shown to mediate sterol/PtdIns(4)$P$ exchange between the ER and Golgi[13]. ORP5 and ORP8 share ~80% sequence identity with each other and are unique members of the ORP family as they lack the FFAT (ER targeting) motif; instead they are the only ORP members with a single C-terminal transmembrane domain (TMD). ORP5 and ORP8 possess the structural features of a lipid transporter: an ER anchor (TMD), a membrane targeting pleckstrin homology (PH) domain and a lipid-binding module (ORD, for OSBP-related domain). These structural features suggest that ORP5 and ORP8, like OSBP, may also be involved in lipid transport at the MCSs between the ER and other cellular membranes.

Indeed, ORP5 and ORP8, and their yeast counterparts Osh6p and Osh7p, have recently been reported to mediate the counter transport of PtdIns(4)$P$/phosphatidylserine (PtdSer) between the ER and plasma membrane (PM): ORP5/8 was shown to transfer PtdSer from the ER to PM, coupled with backward transport of PtdIns(4)$P$ from PM to the ER, resulting in its hydrolysis by the phosphatase Sac1[14–16]. This cycle of forward and backward trafficking of PtdSer and PtdIns(4)$P$ has been shown to help maintain PtdIns(4)$P$ and PtdSer levels in the PM. For instance, the level of PtdSer on the PM was increased while PtdIns(4)$P$ decreased upon ORP5 overexpression[15]. Critical to the transport function of ORP5 and ORP8 is their recruitment to the ER–PM MCSs. It was suggested that the binding of PtdIns(4)$P$ by the PH domain of ORP5/ORP8 alone is sufficient for their distribution to the ER–PM junctions. This finding is based primarily on the observation that overexpression of phosphatidylinositol-4-kinase IIIα, which mediates PtdIns(4)$P$ synthesis, led to increased cortical pool of ORP5, ORP8L, and ORP8S[15].

Here thermodynamic analyses of ORP5 and ORP8 PH domain interactions with phosphoinositides (PtdIns$P$s) clearly demonstrate the preferential binding to PtdIns(3,4,5)$P_3$, PtdIns(4,5)$P_2$, PtdIns(3,4)$P_2$, as well as the late endosome concentrated PtdIns(3,5)$P_2$, but not to PtdIns(4)$P$. The crystal structure of the ORP8 PH domain coupled with mutagenesis, structural and sequence examination provide an explanation for why this PH domain specifically binds di- and tri-phosphorylated PtdIns$P$s rather than PtdIns(4)$P$ as previously suggested[15]. ORP5 and ORP8 ORD domains (ORD5 and ORD8) also bind multiple phosphoinositides. However, in contrast to previous work, we find that knocking down both ORP5 and ORP8 has little effect on PM PtdIns(4)$P$ but dramatically increase the PM levels of PtdIns(4,5)$P_2$. In vitro transport assays also indicate that PtdIns(4,5)$P_2$ is a highly efficient substrate for ORD8 lipid transfer, and a PtdIns(4,5)$P_2$ gradient between donor and acceptor liposomes can greatly facilitate PtdSer transport. Our results confirm the critical importance of ORP5 and ORP8 in ER–PM lipid homeostasis, but show, for the first time, that phosphoinositides other than PtdIns(4)$P$ also serve as co-exchangers for the transport of cargo lipids by ORPs.

## Results

**ORP5 specifically accumulates at the ER–PM junctions.** We have previously shown endogenous ORP5A (isoform A) associates predominantly with ER membranes and the carboxyl terminal transmembrane domain is responsible for ER anchoring[17]. However, overexpressed ORP5A also forms focused puncta around the cell periphery at the mid-cell sections, which appears similar to ER–PM junctions[15]. To investigate the localization of ORP5 and ORP8, we constructed mCherry-tagged ORP5A, naturally occurring ORP8 long (ORP8L) and short (ORP8S) variants that differ by a 42 amino acid stretch at the amino-terminal (Fig. 1a and Supplementary Fig. 1). A genetically engineered tool called MAPPER was employed for selectively monitoring the ER–PM junctions[18]. GFP–MAPPER was co-expressed with mCherry–ORP5A, ORP8L, and ORP8S, respectively, in HeLa cells and examined by confocal microscopy. Overexpressed mCherry–ORP5A labels the cell periphery with punctate structures (Fig. 1b), which almost completely co-localized with the ER–PM junction marker MAPPER. In contrast, both mCherry-tagged isoforms of ORP8 demonstrated only reticular distribution (Fig. 1b). For visualization at the ultrastructural level, GFP–ORP5A expressed in HeLa cells was probed with antisera against GFP for immuno-gold labeling or detected by co-expression of a GFP-binding protein Apex2 construct[19]. Immunoelectron microscopy (EM) on frozen sections revealed the presence of gold-labeled GFP–ORP5A domains at sites of close apposition between the ER and the PM (Fig. 1c). These domains were also clearly visualized using the combination of GFP–ORP5A and GFP-binding protein–Apex2[19] that revealed distinct areas of electron dense staining (Fig. 1d). ORP8 isoforms mostly reside at the reticular ER, in particular ORP8L, which is possibly caused by the negatively charged residues within the first 42 amino acids of ORP8L[15].

**PH domain is required for ORP5 targeting to ER–PM junctions.** ORP5 and ORP8 are tail-anchored ER membrane proteins harboring an N-terminal lipid-binding PH domain. To test if the PH domain is required for trapping ORP5A at the ER–PM junctions, we compared a naturally occurring ORP5 variant, ORP5B where the PH domain is truncated (Fig. 1a). When GFP–ORP5B is expressed in HeLa cells and analyzed by confocal microscopy, its fluorescence pattern is reminiscent of endogenous ORP5[17], mCherry–ORP8L, and mCherry–ORP8S (Fig. 1b, e). Co-localization of GFP–ORP5B with the ER marker (DsRed-ER) shows its enrichment in the ER (Fig. 1e). In contrast, GFP–ORP5A staining shows predominant cortical accumulation, overlapping clearly with the ER–PM junction marker MAPPER (Fig. 1b). Such a reticular accumulation of GFP–ORP5B highlights the importance of the PH domain in recruiting GFP–ORP5A to the ER–PM contact sites (Fig. 1e). This is further confirmed by expression of a GFP–ORP5A construct lacking the PH domain: GFP–ORP5AΔPH is primarily distributed to the reticular ER (Fig. 1e). Moreover, systematic truncation and mutagenesis analyses of the N-terminal region of ORP5A show that the coiled coil domain preceding the PH domain (Supplementary Fig. 1) is critical for the PM tethering (Supplementary Fig. 2). Notably, a single amino acid change (L99A) abolished the targeting of ORP5A to ER–PM contact site. Altogether, these data establish that while the transmembrane domain (TM) of ORP5 maintains anchoring to the ER, the N-terminal coiled coil region together with the PH domain mediates ORP5 tethering to the PM.

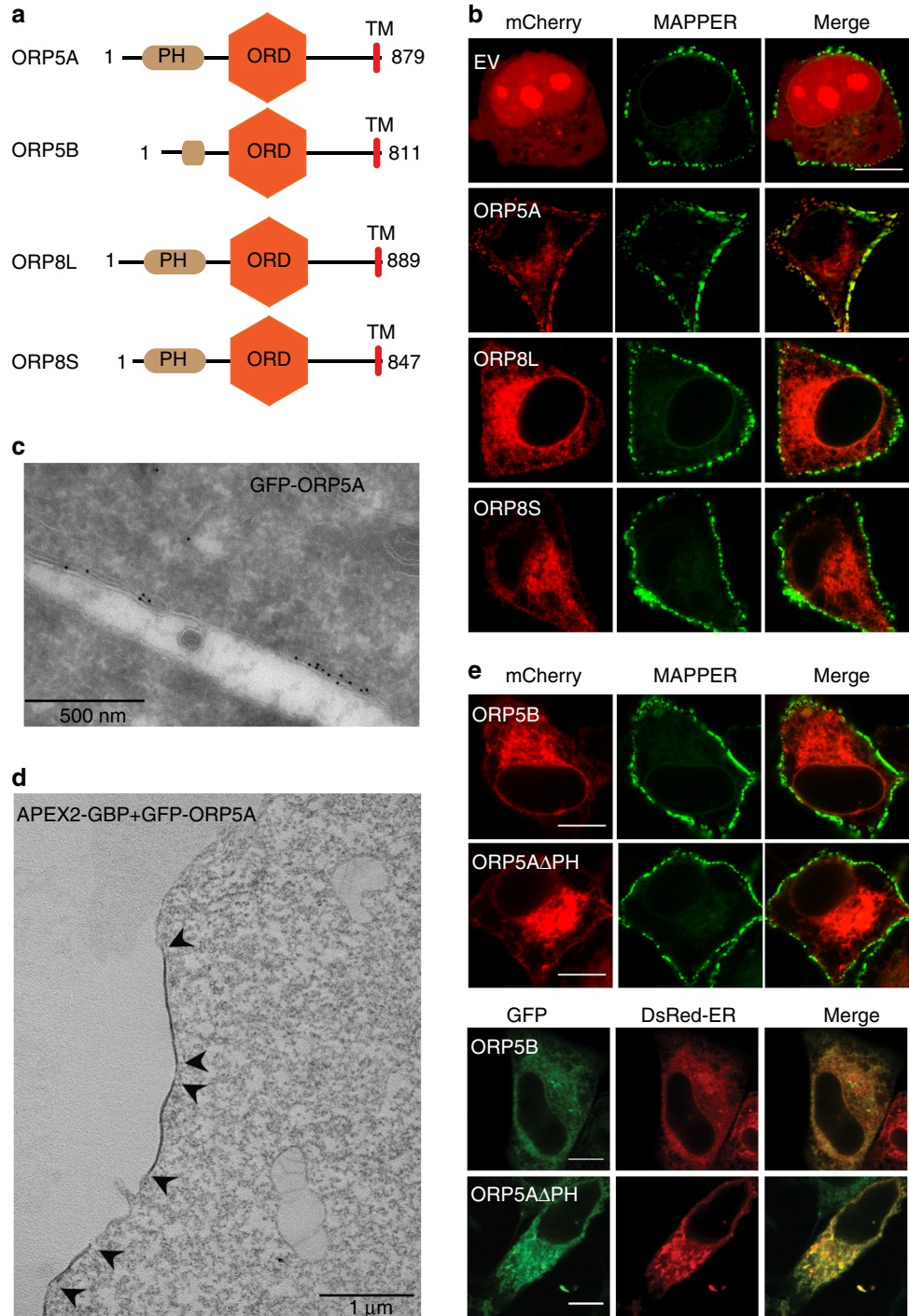

**Fig. 1** Recruitment of ER-anchored ORP5 to the ER–PM contact sites. **a** Schematic representation of the splice variants of human ORP5 and ORP8.
**b** Co-localization of mCherry empty vector (EV) or mCherry-tagged ORP5A, ORP8 long form (ORP8L), and the shorter variant of ORP8 (ORP8S) with
MAPPER (ER–PM junction marker protein) in HeLa cells. *Bar* = 10 µm. **c** Immunoelectron microscopy of HeLa cells expressing GFP-ORP5A highlighting the
labeling of electron dense gold particles at close appositions between the ER and PM. Primary antibody against GFP was used. *Bar* = 500 nm. **d** EM imaging
of HeLa cells expressing APEX-ORP5A, *arrows* highlighting electron dense domain due to ORP5A labeling at the ER–PM junction. *Bar* = 1 µm.
**e** Co-localization of mCherry-tagged ORP5B (variant with a truncated PH domain) and PH domain null ORP5A with MAPPER, and GFP-tagged ORP5B and
ORP5A PH domain deletion construct with DsRed-ER marker demonstrating the importance of PH domain for ER–PM junctional recruitment. *Bar* = 10 µm

**PtdInsPs control ORP5A recruitment to ER–PM junctions.**
The PH domains of OSBP/ORP family members are known to
interact with PtdIns(4)P for tethering to PtdIns(4)P-enriched
membranes (e.g., the Golgi apparatus). A recent study of ORP5
and ORP8 proteins posits that PH–PtdIns(4)P binding serves as a
determinant in the formation of ER–PM junctions[15]. To

characterize the specificity and affinity of ORP5 and ORP8 PH
domains toward PtdIns(4)P, we employed isothermal titration
calorimetry (ITC) using purified ORP5/8 PH domains and short
acyl chain water-soluble phosphoinositide isoforms. Remarkably,
both ORP5 and ORP8 PH domains demonstrate clear preference
for PM abundant di- and tri-phosphorylated PtdInsPs (often

**Table 1 Thermodynamic parameters for the binding of PtdInsPs with ORP5 and ORP8 by ITC[a]**

| Protein | PtdInsP lipid[b] | $K_d$ (μM) | $\Delta H$ (kcal/mol) | $T\Delta S$ (kcal/mol) | $\Delta G$ (kcal/mol) | N |
|---|---|---|---|---|---|---|
| ORP5 PH | PtdInsP | NB[c] | | | | |
| | PtdIns(3)P | NB[c] | | | | |
| | PtdIns(4)P | NB[c] | | | | |
| | PtdIns(5)P | NB[c] | | | | |
| | PtdIns(3,4)$P_2$ | 16.3 ± 0.4 | −0.7 ± 0.4 | 5.8 ± 0.4 | −6.5 ± 0.0 | 1.0 ± 0.0 |
| | PtdIns(3,5)$P_2$ | 25.5 ± 0.5 | −4.6 ± 5.1 | 1.6 ± 4.9 | −6.2 ± 0.1 | 0.9 ± 0.1 |
| | PtdIns(4,5)$P_2$ | 7.3 ± 0.5 | −0.6 ± 0.1 | 6.1 ± 0.3 | −6.7 ± 0.2 | 1.0 ± 0.0 |
| | PtdIns(3,4,5)$P_3$ | 5.3 ± 2.4 | −0.15 ± 0.0 | 7.1 ± 0.2 | −7.2 ± 0.3 | 1.1 ± 0.1 |
| ORP8 PH | PtdInsP | NB[c] | | | | |
| | PtdIns(3)P | NB[c] | | | | |
| | PtdIns(4)P | NB[c] | | | | |
| | PtdIns(5)P | NB[c] | | | | |
| | PtdIns(3,4)$P_2$ | 2.9 ± 0.1 | −3.0 ± 0.1 | 4.5 ± 0.1 | −7.5 ± 0.0 | 1.0 ± 0.0 |
| | PtdIns(3,5)$P_2$ | 13 ± 1.4 | −0.9 ± 0.1 | 5.7 ± 0.1 | −6.6 ± 0.0 | 1.0 ± 0.1 |
| | PtdIns(4,5)$P_2$ | 5.5 ± 1.2 | −4.4 ± 0.1 | 2.7 ± 0.1 | −7.2 ± 0.1 | 1.0 ± 0.0 |
| | PtdIns(3,4,5)$P_3$ | 11.8 ± 3.2 | −2.9 ± 1.2 | 3.7 ± 1.1 | −6.7 ± 0.1 | 1.0 ± 0.1 |
| ORP5 ORD | PtdInsP | NB[c] | | | | |
| | PtdIns(3)P | 9.0 ± 2.7 | −3.1 ± 0.6 | 3.8 ± 0.4 | −6.9 ± 0.2 | 0.9 ± 0.0 |
| | PtdIns(4)P | 6.7 ± 3.3 | −4.2 ± 0.9 | 2.8 ± 1.2 | −7.1 ± 0.3 | 1.0 ± 0.0 |
| | PtdIns(5)P | 26.8 ± 1.9 | −3.0 ± 1.1 | 3.2 ± 1.1 | −6.2 ± 0.0 | 1.0 ± 0.0 |
| | PtdIns(3,4)$P_2$ | 0.8 ± 0.8 | −7.0 ± 0.4 | 1.5 ± 1.2 | −8.5 ± 0.7 | 1.0 ± 0.0 |
| | PtdIns(3,5)$P_2$ | 5.9 ± 3.9 | −5.1 ± 0.5 | 2.0 ± 0.9 | −7.2 ± 0.4 | 1.0 ± 0.1 |
| | PtdIns(4,5)$P_2$ | 6.7 ± 1.5 | −4.2 ± 0.7 | 2.8 ± 0.8 | −7.0 ± 0.1 | 1.0 ± 0.0 |
| | PtdIns(3,4,5)$P_3$ | 1.1 ± 0.3 | −2.2 ± 0.6 | 4.7 ± 0.8 | −6.9 ± 0.2 | 1.1 ± 0.1 |
| ORP8 ORD | PtdInsP | NB[c] | | | | |
| | PtdIns(3)P | 1.7 ± 0.8 | −3.9 ± 2.5 | 1.5 ± 0.8 | −5.3 ± 3.3 | 1.1 ± 0.0 |
| | PtdIns(4)P | 2.3 ± 0.3 | −10.6 ± 7.0 | −2.9 ± 6.9 | −7.7 ± 0.1 | 1.0 ± 0.0 |
| | PtdIns(5)P | 5.9 ± 4.1 | −8.6 ± 0.3 | −1.4 ± 0.1 | −7.2 ± 0.4 | 1.0 ± 0.1 |
| | PtdIns(3,4)$P_2$ | 2.5 ± 0.6 | −14.4 ± 1.6 | −6.7 ± 1.4 | −7.6 ± 0.1 | 1.0 ± 0.0 |
| | PtdIns(3,5)$P_2$ | 3.6 ± 1.9 | −5.9 ± 0.5 | 1.5 ± 0.7 | −7.4 ± 1.2 | 1.1 ± 0.0 |
| | PtdIns(4,5)$P_2$ | 5.3 ± 0.4 | −3.8 ± 0.2 | 3.1 ± 0.5 | −6.9 ± 0.3 | 1.1 ± 0.1 |
| | PtdIns(3,4,5)$P_3$ | 6.0 ± 4.2 | −12.6 ± 2.2 | −5.4 ± 2.7 | −7.2 ± 4.5 | 1.1 ± 0.0 |

[a]Each experiment was performed three times. All measurements are given as average. Errors show standard deviation (s.d.)
[b]PtdInsP species incorporated a water-soluble diC8 acyl chain
[c]NB = no binding detectable

referred to as PIP2 and PIP3, respectively) (Table 1). The dissociation constant ($K_d$) for PH–PtdInsP interactions were in the range of 3–30 μM (Table 1) and the binding was enthalpically driven under the experimental conditions. Notably, no binding was observed with mono-phosphorylated PtdInsPs, including PtdIns(4)P (Supplementary Fig. 3). This strongly suggests that the interaction between the ORP5/8 PH domains and PtdIns(4,5)$P_2$ or PtdIns(3,4,5)$P_3$, but not PtdIns(4)P, may be the driving force in the distribution of ORP5/8 to ER–PM MCS.

**Crystal structure of the atypical PH domain of ORP8.** To understand the structural principles of ER–PM tethering by ORP5/8, the structure of the ORP8 PH domain was determined by X-ray crystallography (Fig. 2a, Supplementary Figs. 4 and 12; and Table 2). The ORP8 PH domain exhibits the conventional core composed of seven β-strands (β1: Val150–Arg158, β2: Thr165–Lys172, β3: Val175–Tyr180, β4: Trp183–Leu188, β5: Glu192–Glu195, β6: Cys204–Phe208, and β7: Tyr237–Arg241) capped by a α-helix (Glu245–Ala258).

Comparison of ORP8 PH domain to the PtdIns(4)P-binding PH domains of LTPs such as Osh3p[20] and Cert[21] as well as the distantly related Ins(1,4,5)$P_3$-binding ARHGAP9 PH domain[22] shows that ORP8 PH domain superposes well with an average r.m.s.d. of 1.5 Å over 85 Cα atoms. A distinctive feature that stands out, however, is the presence of an unusually long 25 amino acid stretch forming an extended β6–β7 loop (His209–Ser236) (Fig. 2b). Comparative analysis of electrostatic surface of ORP8 PH domain with Osh3p PH domain reveals a distinct positively charged cavity, representing a putative PtdInsP-binding region (Supplementary Fig. 5A). The analogous surface on the Osh3p PH domain has a neutral charge suggesting that ORP5 and ORP8 PH domains may possess a distinct PtdInsP-binding mechanism.

**A non-canonical PtdIns P-binding site of ORP5/8 PH domain.** Close inspection of the distinct positively charged cavity of ORP8 PH domain shows the amino acids forming the basic patch are lined on the β1–β2 and β5–β6 loops (non-canonical mode) (Fig. 2c and Supplementary Fig. 5A, B). Overlay of ORP8, Osh3p[20], Cert[21], ARNO[23], and ARHGAP9[22] PH domain shows clearly that ARHGAP9 bound Ins(1,4,5)$P_3$ (non-canonical) is oriented between the β1–β2 and β5–β6 loops of ORP8 PH domain (Fig. 2c). The PtdIns(4,5)$P_2$ shown in Fig. 2c is modeled by superposing Ins(1,4,5)$P_3$ bound to AHGAP9. The electrostatic surface model constructed by superimposing ARHGAP9 PH[22] on to ORP8 PH highlights that PtdIns(4,5)$P_2$ is arranged in the putative non-classical lipid-binding basic pocket (Fig. 2c).

Structure-based sequence alignments of related PH domain containing LTPs ORP5, ORP8, OSBP, Cert, and Osh3p establish that the putative PtdInsP residues (red inverted triangles) in ORP5 and ORP8 PH domain are not conserved in OSBP, Cert, and Osh3p (Fig. 2d). Notably the side chains (green triangles) mediating canonical PtdIns(4)P interaction with OSBP, Cert, and Osh3p are missing in ORP5 and ORP8. To confirm the importance of the non-canonical basic surface for PtdInsP binding, we designed point mutations R136Q, R179Q, and R158Q, R201Q in the ORP5 and ORP8 PH domains, respectively. Both the point mutations in ORP5 and ORP8 PH domains

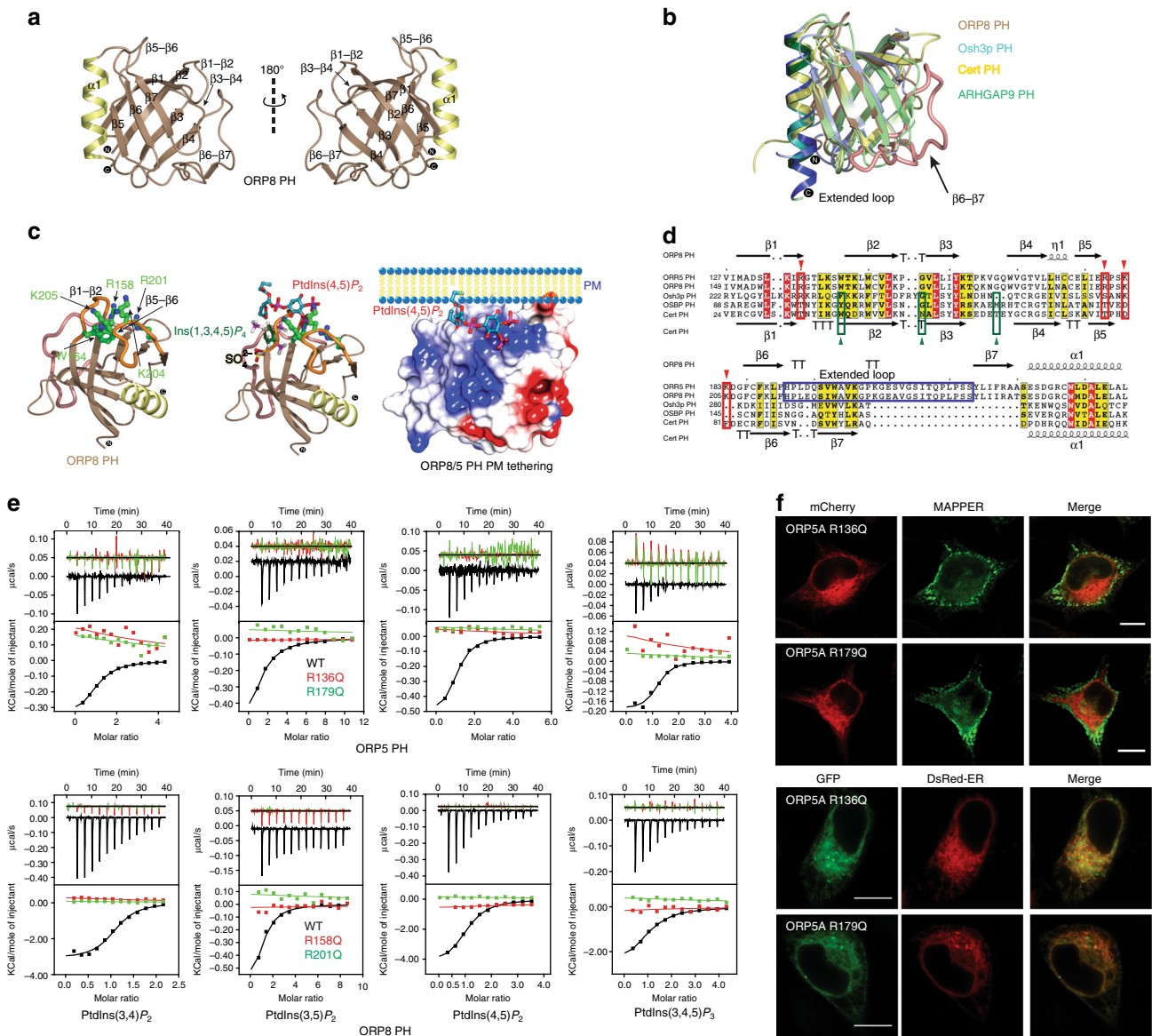

**Fig. 2** Non-canonical PH domain–PtdInsP association is indispensable for translocation of ORP5 to ER–PM junctions. **a** Cartoon representation of the crystal structure of ORP8 PH domain. **b** Ribbon representation of the superposition of ORP8 PH (*brown*), Osh3p PH (*cyan*; PDB id: 4IAP)[20], Cert PH (*yellow*; PDB id: 2RSG)[20], and ARHGAP9 PH (*green*; PDB id: 2P0D)[20]. **c** Model of ORP8 PH domain constructed by superimposing PH domains of ARHGAP9[20], Cert[20], and ARNO[20], highlights the putative PtdInsP-binding site (between β1–β2 and β5–β6). The electrostatic surface representation shows the presence of a positively charged cleft presented by the β1–β2 and β5–β6 loops for PtdInsP binding. Electrostatic potential rendered surface was computed in ccp4mg[45], negatively charged surfaces are shown in *red*, whereas positively charged surfaces are *blue* in color, *colors* are contoured from −0.5 V to +0.5. **d** A combined sequence alignment and secondary structure comparison of the PH domain of ORP5, ORP8, Osh3p, OSBP, and Cert. Secondary structure elements for ORP8 and Cert PH derived from the crystal structure are indicated *above* and *below* the alignment, respectively. Alignments were made with ESPript 2.2 (http://espript.ibcp.fr/ESPript/ESPript/)[46]. *Red inverted triangles* indicate the positively charged amino acids constituting the basic patch on the PH module of ORP8 and ORP5, which are absent in the other PH domains. *Green triangles* indicate the amino acids mediating PtdIns(4)P binding on the PH domains of Osh3p, Cert, and OSBP, which are absent in ORP5 and ORP8 PH domains. **e** The binding of ORP5 and ORP8 PH domain to PtdInsPs was measured by ITC. See Table 1 for a complete list of results. The binding of ORP5 R136Q, R179Q, ORP8 R158Q, R201Q, and WT ORP5 and ORP8 PH domains are shown in *red*, *green*, and *black*, respectively. Experiments were performed at 25 °C using 25 μM protein and 500 μM PtdInsPs. *Top panels* show raw data and *bottom panels* show integrated normalized data. **f** Co-localization of GFP and mCherry-tagged WT ORP5A and mutants (R136Q, R179Q) with MAPPER and DsRed-ER in HeLa cells. The GFP-fused ORP5A mutants (R136Q, R179Q) mainly overlap with DsRed-ER, but not MAPPER, suggesting that these mutants lose cortical ER localization. *Bar* = 10 μm

completely abolished the binding to PtdInsPs (Fig. 2e), confirming that ORP5 and ORP8 PH domain follows a non-canonical PtdInsP-binding regime.

To test the PtdInsP dependency of ORP5A localization to the ER–PM junctions, structure-based point mutants R136Q and R179Q in the GFP-tagged full-length ORP5A were expressed in

HeLa cells and the cellular localization was analyzed using confocal microscopy. While the wild-type (WT) GFP–ORP5A selectively labels the ER–PM junctions and co-localizes well with MAPPER (Fig. 1b), both GFP–ORP5A(R136Q) and GFP–ORP5A(R179Q) completely lost the cortical accumulation (Fig. 2f).

**Table 2 Summary of crystallographic structure determination statistics[a]**

|  | ORP8 (149–265) | ORP8 (149–265) co-crystallized with IP6 |
|---|---|---|
| *Data collection* |  |  |
| Space group | P 65 2 2 | P 1 21 1 |
| Unit cell dimensions (a, b, c; α, β, γ) | 57.1 Å, 57.1 Å, 154.9 Å, 90°, 90°, 120° | 52.1 Å, 66.3 Å, 79.5 Å, 90°, 94.6°, 90° |
| Total reflections | 134158 (7609) | 271018 (38350) |
| Wavelength (Å) | 0.9537 | 0.9537 |
| Resolution range (Å) | 51.65–2.27 (2.16) | 79.29–2.1 (1.98) |
| Mean I/sigma (σI) | 18.4 (1.9) | 9.7 (2.8) |
| R-merge | 0.114 (1.352) | 0.10 (0.527) |
| Unique reflections | 7941 (756) | 37677 (5410) |
| Multiplicity | 16.9 (10.1) | 7.2 (7.1) |
| Mn(I) half-set correlation CC (1/2) | 0.999 (0.594) | 0.99 (0.95) |
| Completeness (%) | 91.0 (63.0) | 99.2 (98.1) |
| Wilson B-factor | 35.9 | 30.1 |
| *Refinement* |  |  |
| R-work | 0.22 (0.25) | 0.22 (0.29) |
| R-free | 0.27 (0.33) | 0.25 (0.34) |
| Resolution range (Å) | 47.09–2.16 | 40.92–1.98 |
| Number of atoms | 955 | 3931 |
| Protein atoms | 920 | 3739 |
| RMS (bonds) | 0.014 | 0.012 |
| RMS (angles) | 1.443 | 1.245 |
| Ramachandran favored (%) | 100 | 99.57 |
| Ramachandran outliers (%) | 0 | 0 |
| Average B-factor | 39.27 | 27.15 |

[a]Highest resolution shell is shown in parentheses

**PtdIns(4,5)$P_2$-dependent ORP5/8 ER–PM junctional recruitment**. Our lipid binding and structural data suggest that ORP5A accumulation at the ER–PM junction is potentially modulated by PtdInsPs such as the PM-enriched PtdIns(4,5)$P_2$, but not PtdIns(4)$P$ as previously reported[15]. To test this, we employed a genetically encoded rapamycin inducible pseudojanin (PJ). PJ is a fusion of PtdIns(4)$P$ phosphatatse (Sac1) and PtdIns(4,5)$P_2$ phosphatase inositol polyphosphate-5-phosphatase E (INPP5E) with the FK506-binding protein (FKBP) domain, which can be specifically recruited to the PM in cells expressing the FKBP-rapamycin-binding (FRB)-tagged PM marker by addition of rapamycin[24]. Thus rapamycin gives rise to PM-specific depletion of PtdIns(4)$P$ and/or PtdIns(4,5)$P_2$ pools (Fig. 3a, b), and PJ mutants PJ–Sac1 (inactive INPP5E) and PJ-INPP5E (inactive Sac1) cause depletion of PtdIns(4)$P$ and PtdIns(4,5)$P_2$, respectively[24]. GFP–ORP5A was co-transfected in HeLa cells with the PJ plasmids and analyzed by confocal microscopy. While the PJ-Dead control had no effect on ORP5A localization to the junctions, activation of PJ to the PM with rapamycin renders the GFP–ORP5A completely ER localized (Fig. 3a. b). Importantly, PM recruitment of PtdIns(4,5)$P_2$ phosphatase (PJ-INPP5E) alone triggers dramatic relocation of GFP–ORP5A from the contact sites to the reticular ER, whereas depletion of PtdIns(4)$P$ by PJ-Sac had no effect (Fig. 3a, b). These observations suggest that localization of GFP–ORP5A to ER–PM junctions is regulated specifically by PtdIns(4,5)$P_2$.

Since GFP–ORP8 does not localize to ER–PM junction (Fig. 1b), we investigated whether elevated PtdIns(4,5)$P_2$ at the PM could drive ORP8 to the ER–PM junctions. Phosphatidylinositol-4-phosphate 5-kinase type-1 beta (PIP5K1b) selectively phosphorylates PtdIns(4)$P$ to produce PtdIns(4,5)$P_2$ at the PM (Supplementary Fig. 6A–C). Upon co-expression of PIP5K1b and mCherry–ORP8L, we observed dramatic recruitment of ORP8L to the ER–PM MCS (Fig. 3c, d and Supplementary Fig. 6D). On the contrary, expression of PIP5K1b with mCherry–ORP8L(R158Q) does not result in translocation of the mutant ORP8L to the ER–PM junctions (Fig. 3c, d and Supplementary Fig. 6D). These results confirm that the targeting of ORP5/8 to the ER–PM junctions is dependent on PH domain binding to PtdIns(4,5)$P_2$.

**ORP5/8 bind, extract, and transport various PtdIns $P$s**. Both ORP5 and OPR8 possess the hallmark lipid transfer ORD domain, and the lipid cargo for ORD5 and ORD8 still remains to be firmly established. To identify which phospholipids, particularly PtdInsPs, serve as cargo for ORD5 and ORD8, we successfully purified recombinant ORD5 and ORD8 in bacteria (*Escherichia coli*) (Supplementary Fig. 7). By utilizing short acyl chain water-soluble phosphoinositide derivatives, we characterized the lipid-binding attributes of ORD5 and ORD8 by ITC. Both ORD5 and ORD8 bound all the PtdInsP, including PtdIns(4)$P$, but not PtdIns. ORDs of ORP5 and ORP8 associated with other PtdInsPs with similar affinity to PtdIns(4)$P$, which is thought to be a cargo lipid for ORP5 and ORP8 (Fig. 4a, Table 1, and Supplementary Fig. 8)[15].

Osh6p, the closest homolog of ORP5/8, has been shown to bind, extract, and transport PtdIns(4)$P$[6]. To gain mechanistic understating of how ORP5/8 binds by PtdInsPs, we tried crystallizing ORD5/8 in complex with various lipids. While ORD8 crystals grew, they did not diffract. Therefore, in the absence of a crystal structure we performed homology modeling for ORD5 and ORD8 using Osh6p as the template. The ORD5/8 model overlays Osh6p structure well with an r.m.s.d. of 0.9 Å over 328 amino acids (Fig. 4b). The overlay of Osh6p–PtdIns(4)$P$ complex with homology model of ORD5/8 demonstrates the presence of a conserved cargo-binding site. Osh6p amino acids coordinating interaction with PtdIns(4)$P$ are conserved in both ORD5 and ORD8 (Fig. 4b). Notably, mutating the two conserved histidine residues in ORP5 (H478A, H479A) or ORP8 (H514A, H515A) abolished the binding with all the PtdInsP species including PtdIns(4)$P$ and PtdIns(4,5)$P_2$ (Fig. 4a).

The ITC-binding data and our modeling suggest that in addition to PtdIns(4)$P$, other PtdInsPs could be extracted from donor membranes and transported to the acceptor membranes by ORD5/8. To investigate this, we first determined whether purified ORD8 domain (Supplementary Fig. 7C) extracted PtdIns(4)$P$ or PtdIns(4,5)$P_2$ from model membranes. We used a PtdInsPs sensor, NBD–PH$_{FAPP}$ (PH domain of the four-phosphate-adapter protein-1)[14, 25], whose (7-nitrobenz-2-oxa-1, 3-diazol) NBD fluorophore fluoresces strongly in the membrane-bound form, but not in the soluble form. Although PH–FAPP1 was known to have high affinity for PtdIns(4)$P$, our data clearly show that NBD–PH$_{FAPP}$ associates strongly with membranes doped with various PtdInsP species except PtdIns (Supplementary Fig. 9A, B). In the absence of an ORD domain, NBD–PH$_{FAPP}$ fluorescence was very strong because the fluorophore is fully bound to PtdInsPs-containing liposomes. Adding purified Osh6p or ORD8 reduced NBD fluorescence, indicating that PtdInsP has been extracted from liposomes by these proteins (Fig. 5a). Importantly, ORD8 efficiently extracted PtdIns(4,5)$P_2$, in addition to PtdIns(4)$P$. In control experiments, WT-ORD8 efficiently extracted PtdIns(4)$P$/PI(4,5)$P_2$, whereas ORP8(H514A, H515A), deficient in binding PtdIns(4)$P$/PtdIns(4,5)$P_2$, exhibited much reduced extraction capacity (Fig. 5a). Additionally, ORD8 can extract phospholipids of varying acyl chains as well as brain PtdIns(4)$P$,

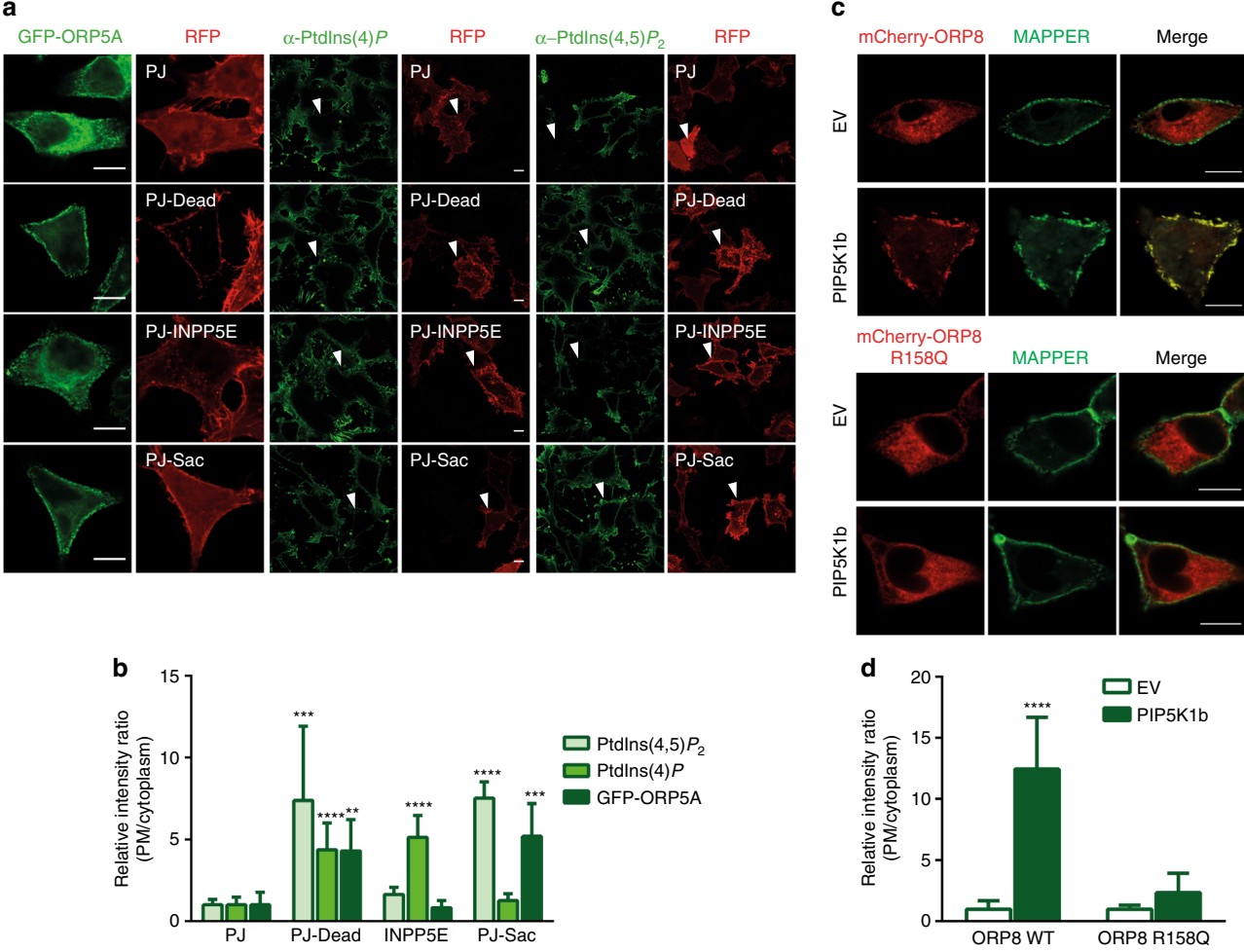

**Fig. 3** Distribution of ORP5 to the ER–PM junctions is PtdIns(4,5)$P_2$ dependent. **a** Effect of PJ (a fusion protein of inositol polyphosphate-5-phosphatase E (INPP5E) and the *S. cerevisiae* Sac1 phosphatase), PJ-Sac (with inactivated INPP5E domain), or PJ-INPP5E (lacking the sac domain) on GFP–ORP5A, PtdIns(4)$P$ and PtdIns(4,5)$P_2$ distribution after PM recruitment for 5 min with 1 μM rapamycin. **b** Quantitation of PM/cytoplasm intensity ratio of GFP–ORP5A, PtdIns(4)$P$ and PtdIns(4,5)$P_2$ in relation to the effect of PJ and its mutants as in **a** (mean + s.d.; ****$P < 0.0001$; ***$P < 0.001$; **$P < 0.01$, one-way ANOVA, $n = 8$ - 15 cells). **c** Co-localization of mCherry–ORP8L and mCherry–ORP8L R158Q with MAPPER upon PIP5K1b overexpression. **d** Quantitation of PM/cytoplasm intensity ratio of mCherry–ORP8L and -ORP8L R158Q as shown in **c** (mean + s.d.; ****$P < 0.0001$, *t* test, $n = 6$ cells). *Bar* = 10 μm

PtdIns(4,5)$P_2$ (the predominant form is C18:0/C20:4), suggesting that ORP8 could extract and transport multiple PtdIns$P$ species (Supplementary Fig. 9C).

We therefore examined the role of ORD8 in PtdIns$P$ transport in vitro using an assay as described[14]. NBD–PH$_{FAPP}$ was mixed with two liposome populations: the donor liposome (L$_A$) contains DOPC, 4% PtdIns$P$s, and 2% rhodamine phosphatidylethanolamine (Rhod-PE); the acceptor liposome (L$_B$) contains DOPC alone, or with 5% PtdSer (Fig. 5b). The fluorescence of NBD–PH$_{FAPP}$, when bound to L$_A$ liposomes through PtdIns$P$s, was quenched due to fluorescence resonance energy transfer with Rhod-PE. Adding ORD8/Osh6p would result in dequenching if the proteins can transport PtdIns$P$s to L$_B$ liposomes (Fig. 5b–e). In control experiments, Osh6p transports C16:0/C16:0 PtdIns(4)$P$ and brain PtdIns(4,5)$P_2$ efficiently, but not brain PtdIns(4)$P$ (Fig. 5e and Supplementary Fig. 9D, E). Likewise, ORD8 can transfer C16:0/C16:0 PtdIns(4)$P$ and brain PtdIns(4,5)$P_2$ efficiently, but not brain PtdIns(4)$P$ (Fig. 5c–e and Supplementary Fig. 9F). Notably, ORD8 and Osh6p transported brain PtdIns(4,5)$P_2$ much more efficiently than PtdIns(4)$P$ as reflected by the initial transport rate ($v_0$) (Fig. 5e and Supplementary Fig. 9F). Moreover, our results show that having PtdSer in L$_B$ did not dramatically enhance ORD8-mediated PtdIns(4,5)$P_2$ and

PtdIns(4)$P$ transport (Fig. 5c–e). ORD8 (H514A, H515A) mutant almost completely ablated PtdIns(4,5)$P_2$ transport (Fig. 5d, e), suggesting that ORD8 adopts a conserved mechanism to transport PtdIns(4,5)$P_2$ to that of Osh6p for PtdIns(4)$P$ transport[14]. Together, these results clearly demonstrate that ORD8 can transfer PtdIns(4,5)$P_2$ more efficiently than PtdIns(4)$P$, and that the acyl chain composition of PtdIns$P$s can greatly impact the rates of PtdIns$P$ transfer by the ORPs.

ORP5 and ORP8 were reported to regulate the level of PtdIns(4)$P$ at the PM by mediating PtdIns(4)$P$ transport from the PM to the ER for hydrolysis[15]. Since our data strongly suggest that ORP5/8 binds and transports other PtdIns$P$s, such as PtdIns(4,5)$P_2$, more efficiently, we sought to investigate if endogenous ORP5/8 may also regulate PtdIns(4,5)$P_2$ at the PM. We first overexpressed ORP5 or ORP5ΔPH in HeLa cells. The amount of PtdIns(4,5)$P_2$ at the PM was significantly reduced upon expressing ORP5, but not ORP5ΔPH as indicated by the fluorescence intensity of PH-PLC–GFP[26], a well-established PtdIns(4,5)$P_2$ marker (Fig. 5f, g). We then knocked down both ORP5 and ORP8 in HeLa cells, and this led to a significant increase of PtdIns(4,5)$P_2$ at the PM (Fig. 5h–j and Supplementary Fig. 11A). These data convincingly highlight the role of both ORP5 and OPR8 in maintaining PtdIns(4,5)$P_2$ homeostasis at the PM.

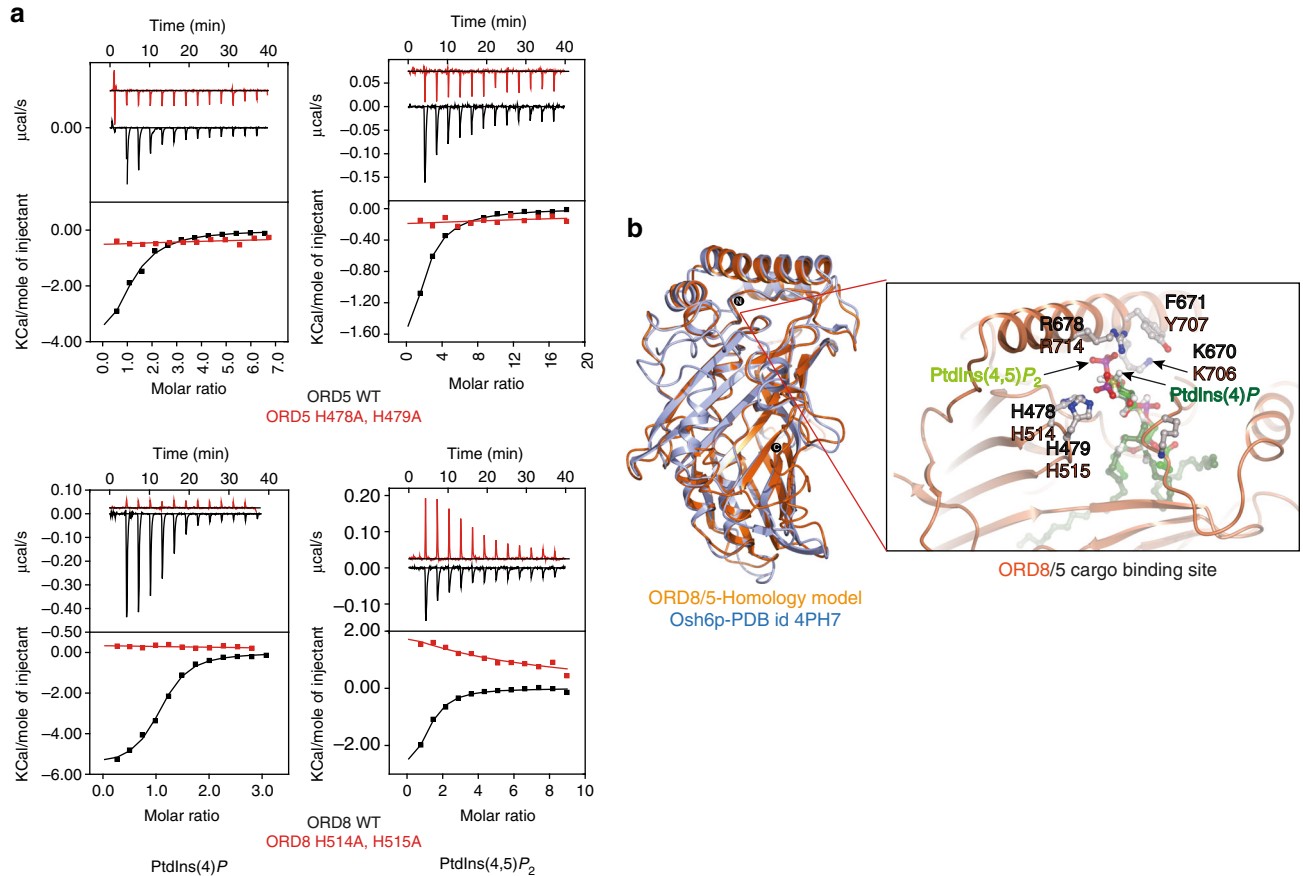

**Fig. 4** ORP5 and ORP8 bind PtdIns*P* species through a conserved binding mechanism. **a** The binding of purified ORP5 and ORP8 ORD domain to water-soluble PtdIns*P* species was measured by ITC. The WT ORD domain of ORP5 and ORP8 binds to all the PtdIns*P*s (Table 1 and Supplementary Fig. 8), including PtdIns(4)*P*. No binding was observed with ORD5 (H478A, H479A) and ORD8 (H514A, H515A) suggesting a conserved cargo-binding mode. Experiments were performed at 25 °C using 25 μM protein in the cell and 500 μM PtdIns*P*s injected from the syringe. *Top panels* show raw data, and *bottom panels* show integrated normalized data. **b** The homology model of human ORD8 and ORD5 (*orange*) is superimposed on the Osh6p (*blue*, PDB id: 4Ph7, 4B2Z)[14, 16] demonstrating their similar overall architectures. The blow up cartoon representation of the putative cargo-binding site in the ORP8/5 ORD is constructed by superposition of the Osh6p–PtdIns(4)*P* complex on the ORD8 homology model. The amino acids that mediate cargo binding in the Osh6p structure are conserved in both ORD8 (*orange*) and ORD5 (*black*)

**A PtdIns(4,5)$P_2$ gradient enhanced PtdSer transport by ORD8.**
ORP5/8 has been reported to counter transport PtdSer and PtdIns(4)*P* between the ER and PM[15]. Importantly, the transport of PtdSer by ORP8 in vitro was significantly augmented by the presence of PtdIns(4)*P* in acceptor liposomes[15]. A PtdIns(4)*P* gradient is also known to be the driving force of PtdSer transport by a yeast ORP5/ORP8 homolog, Osh6p[14]. To assess the effect of other PtdIns*P*s, we carried out in vitro PtdSer transport assays employing a PtdSer-selective fluorescent sensor (NBD-C2$_{Lact}$, based on the lactadherin C2 domain) (Supplementary Fig. 10A, B)[14]. NBD-C2$_{Lact}$ was mixed with two types of liposomes: the donor (L$_A$) contains 5% PtdSer and 2% Rhod-PE and the acceptor (L$_B$) contains DOPC alone or with 4% PtdIns*P*s (Fig. 6a). When bound to L$_A$ liposomes, NBD-C2$_{Lact}$ signal was quenched due to fluorescence resonance energy transfer with Rhod-PE. Adding ORD8 restored NBD-C2$_{Lact}$ emission due to ORD8-mediated PtdSer transfer to L$_B$, followed by NBD-C2$_{Lact}$ binding to L$_B$ liposomes where there is no Rhod-PE (Fig. 6a–c). While WT-ORD8 efficiently transported PtdSer in the presence of brain PtdIns(4,5)$P_2$-containing L$_B$, ORD8 (H514A/H515A) mutant demonstrated much weakened capacity in PtdSer transport (Fig. 6b, c). The L69D mutation in Osh6p was shown to disrupt PtdSer extraction and transport[16], but an analogous L425D mutation in ORP8 surprisingly had little effect on PtdSer

transport (Fig. 6b, c), suggesting other residues may be coordinating association with PtdSer. There are three important observations from this assay. First, a PtdIns*P* gradient can dramatically enhance the transport efficiency of PtdSer by ORD8 (Fig. 6b, c). The presence of 4% PtdIns(4,5)$P_2$ in L$_B$ increased the initial transport rate ($v_0$) of PtdSer by ORD8 from 0.51 to 6.96 PtdSer/min per ORD8 ($v_0$ calculated from the signal converted into PtdSer amount in L$_B$ liposomes) (Fig. 6c). Second, PtdIns(4,5)$P_2$ had a stronger stimulatory effect on PtdSer transport mediated by ORD8 than other PtdIns*P*s, including PtdIns(4)*P*, suggesting PtdIns(4,5)$P_2$ may serve as a preferred and more efficient counter exchanger with PtdSer at ER–PM contact sites (Fig. 6b, c). Third, in this assay system, PtdIns(3)*P* also had a strong stimulatory effect on PS transport, suggesting that ORD8 may utilize multiple PtdIns*P*s for the counter transport of PS (Fig. 6b, c). As a control, Osh6p efficiently transported PtdSer in the presence of C16:/C16:0 PtdIns(4)*P* (Fig. 6c and Supplementary Fig. 10C). However, Osh6p could not transport PtdSer in the presence of brain PtdIns(4)*P* (Supplementary Fig. 10C). These data further suggest that the acyl chain composition of PtdIns*P*s can greatly impact the rates of the counter-transfer process.

Next, we investigated whether ORP5/8 can modulate PtdSer distribution in cells. When both ORP5/8 were knocked down as described in Fig. 5g, the PM intensity of a PtdSer marker Lact-

C2–GFP was significantly decreased (Fig. 6d, e), suggesting defective PtdSer transport from the ER to PM and confirming the role of ORP5/8 in PtdSer metabolism/distribution in cells.

## Discussion

Non-vesicular lipid transfer between the membranes of different organelles is now recognized as a major contributor to membrane homeostasis. OSBP and ORPs/Oshs are major LTPs in eukaryotic cells that mediate the intermembrane transport of lipids. The prevailing paradigm of the transport function of ORPs rests on two core features: (1) Most ORPs operate at membrane contact sites/junctions, and certain ORPs (e.g., OSBP, ORP5/8) can also drive membrane tethering in part through the binding of PtdIns(4)$P$ with their respective PH domains. (2) The forward transport of lipids from the ER (site of synthesis) is often coupled to the reciprocal transport of PtdIns(4)$P$ to the ER, where it is hydrolyzed by the ER phosphatase Sac1. The hydrolysis of PtdIns(4)$P$ is believed to promote the continuous

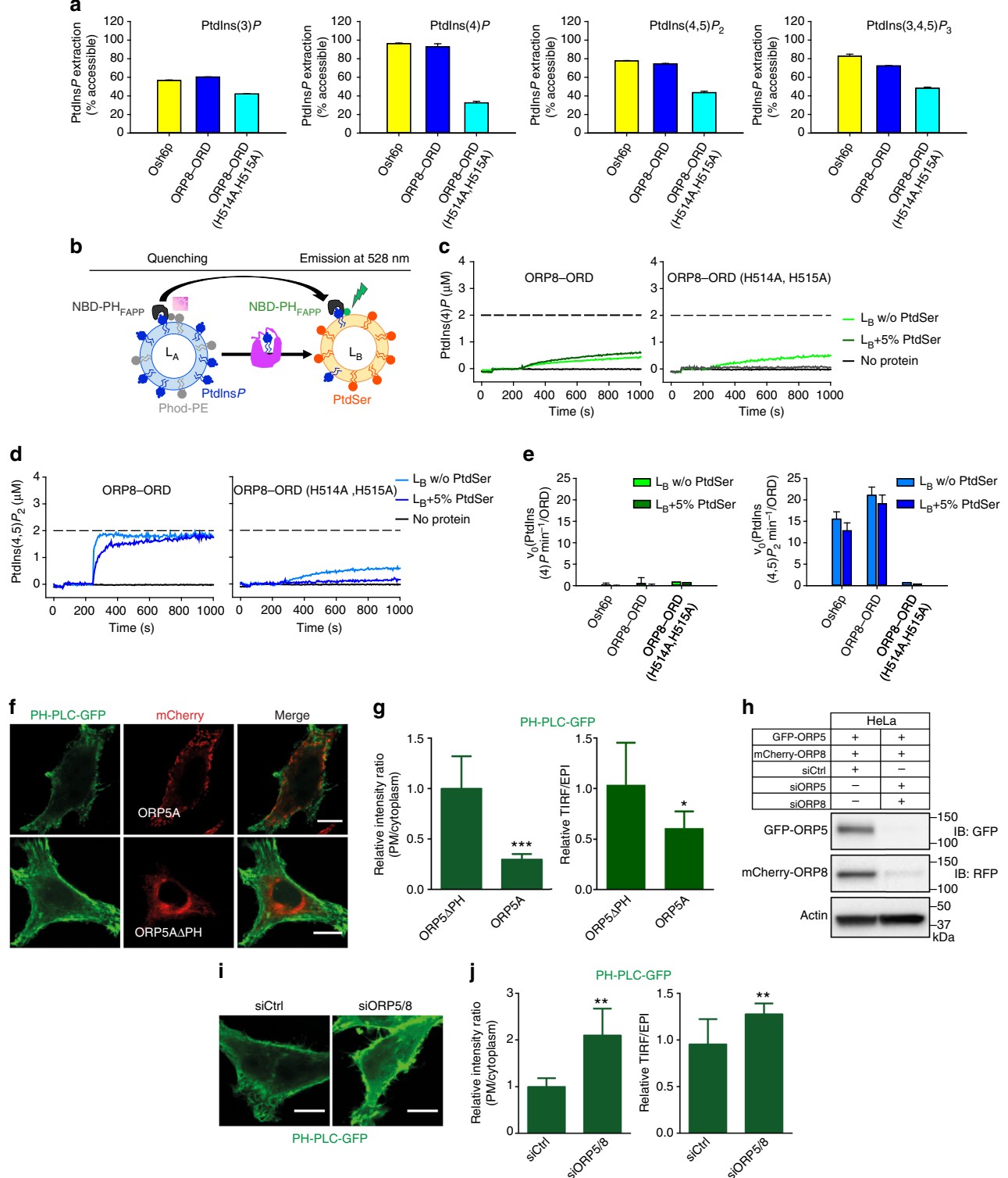

transport of the other lipid cargo against a concentration gradient by the ORPs[10, 13].

Our data herein expands, and in some ways contradicts, this established model of ORP function. First, we show that ORP5/8 can localize to the ER–PM junction through its PH domain that serves as a membrane tether. Our structural and biochemical data strongly support a critical role for PtdIns(4,5)$P_2$, but not PtdIns(4)$P$, in the targeting of ORP5/8 to the ER–PM junction. The PH domain of ORP5/8 shares very poor sequence identity with Osh3 and all the other long ORP family members[20]. While most PH domains (including Osh3, OSBP) utilize the canonical PtdIns$P$-binding regime for PM targeting[20], the PH domain of ORP5/8 possesses an atypical PtdIns$P$-binding site that does not permit PtdIns(4)$P$ binding. The presence of unique PtdIns$P$-binding determinants in ORP5/8 posits that PtdIns(4,5)$P_2$-dependent PM targeting mechanism is exclusive for these two family members. It also challenges the idea that PH–PtdIns(4)$P$ interaction is a general hallmark of yeast and mammalian ORPs for the formation of ER–PM contact sites. Although ours is the first report describing this new PM tethering phenomenon by ORPs, there are other LTPs, such as the extended-synaptotagmins (E-Syts), that utilize the C2 domain–PtdIns(4,5)$P_2$ interaction for tethering the ER to the PM[27].

The PH domain alone is not sufficient for ORP5/OPR8 PM tethering and the positively charged coiled coil preceding the PH domain is also crucial for providing avidity for membrane association. Our systematic truncation and mutagenesis in ORP5 shows mutation of a critical residue in the coiled coil region redistributes OPR5 to the reticular ER. In the case of OSBP, PH-Arf1-GTP association is a crucial factor in ensuring the stability of the ER–Golgi tether[13]. The presence of multiple membrane-binding C2 domains in E-Syts further strengthens the thought that avidity is a vital factor in the maintenance of robust ER–PM junctions[5, 27–29]. Our study has provided a comprehensive understanding of how ORP5 and ORP8 serve as ER–PM tethering molecules. However, we still lack an understanding of what triggers the PM translocation of ORP5/8, integral ER membrane proteins.

Our data also challenge the notion that PtdIns(4)$P$ is the sole ligand for all ORP/Osh proteins. Through systematic PtdIns$P$-binding analysis of ORD5 and ORD8, we have for the first time provided evidence that ORD of ORP5/8 bind all the PtdIns$P$s with similar affinity to the canonical ORP/Osh cargo, PtdIns(4)$P$[10, 13, 15, 20, 30]. Contrary to the current belief that ORP5/8 serve as a PtdIns(4)$P$/PtdSer counter exchanger between ER–PM bilayers[15], we demonstrate the ability of ORP5/8 to transfer PtdIns(4,5)$P_2$ more efficiently in exchange for PtdSer. Notably, PtdIns(4,5)$P_2$ associates with the ORP5/8 cargo-binding tunnel by utilizing conserved molecular determinants[14, 15], suggesting that the cargo-binding pocket has sufficient plasticity/room to accommodate the 5-phosphate of PtdIns(4,5)$P_2$. Importantly, a PtdIns(4,5)$P_2$ gradient can efficiently enhance the transfer of PtdSer by ORD8. Absence of ORP5/8 from cells results in

accumulation of PtdIns(4,5)$P_2$, and loss of PtdSer at the plasma membrane, respectively. Considering the abundance of PtdIns(4,5)$P_2$ in the PM, the physiological function of ORP5/8 is likely to regulate forward transfer of PtdSer coupled with backward transfer of PtdIns(4,5)$P_2$. The fact that the PH domain of ORP5/8 specifically binds PtdIns(4,5)$P_2$, but not PtdIns(4)$P$, for tethering ER and PM lends further support to the involvement of PtdIns(4,5)$P_2$ in ORP5/8-mediated PtdSer transport. Given the relative minute quantity of PtdIns$P$s as compared to PtdSer or cholesterol on the inner leaflet of the PM, it is perhaps favorable that ORP5/8 can utilize multiple PtdIns$P$s for efficient cargo exchange. Moreover, PtdIns(4,5)$P_2$ may enable a more specific and stronger ORD–lipid interaction because it is more negatively charged than PtdIns(4)$P$.

Following extraction and transport by ORP/Osh proteins to the ER, PtdIns(4)$P$ undergoes hydrolysis by ER resident 4-phosphatase, Sac1[10, 13–15, 25, 30]. This activity converts PtdIns(4)$P$ into PtdIns. Another ER-anchored LTP called Nir2 transports the newly generated PtdIns from the ER to the PM in exchange of phosphatidic acid for sustained cellular signaling[31]. It is, however, unclear how PtdIns(4,5)$P_2$ is metabolized after extraction from the PM by ORP5/8. The two phosphate groups of PtdIns(4,5)$P_2$ may be removed by the sequential activity of a 5-phosphatase (e.g., INPP5E) and Sac1, generating PtdIns for forward transport to PM by Nir2 or other PtdIns transfer proteins[32, 33]. Modulation of PtdIns(4,5)$P_2$ levels at the PM is of critical importance for several cellular functions including membrane trafficking and cellular signaling[34]. This puts the onus on LTPs, such as ORP5/8, to exchange PtdIns(4,5)$P_2$ with PtdSer at an optimal rate. We and others have shown that ORP5 and ORP8 physically associate with each other (Supplementary Fig. 11B, C)[15, 35]. Therefore, it is plausible that ORP5 and ORP8 are in a heterodimeric state at the junctions. Such a heterodimeric lipid shuttle at the ER–PM MCS could exponentially increase the rate of lipid transfer when needed.

Recently, there have been several reports demonstrating the dynamicity of cortical ER in forming junctions with almost all other cellular organelles[36]. For instance, ORP1L induces the formation of ER-late endosome (LE) membrane contact site for cholesterol transfer[37, 38]. The fact that ORP5 can regulate endosomal cholesterol redistribution suggests that the ER-anchored ORP5/8 potentially may also act as a tether to create ER–LE junctions[17]. Notably, ORP5/8 PH domain binds to LE-enriched PtdIns(3,5)$P_2$, indicating that ORP5/8 does not exclusively translocate to ER–PM junctions, but can be recruited to various MCSs under certain physiological conditions. In this regard, ORP5 and ORP8 have recently been reported to also localize to the ER–mitochondria contact sites[35]. Another important aspect of ORP5/8 function is the close relationship between cholesterol and PtdSer on the cytoplasmic leaflets of cellular membranes[39], and loss of ORP5/8 may have a profound impact on the distribution of cholesterol, and possibly other lipids. In addition, whether ORP5/8 strictly transports PtdSer

**Fig. 5** ORD8 can extract and transport PtdIns(4,5)$P_2$. **a** Bar graph showing the percentage of PtdIns$P$ extracted from liposomes by Osh6p, ORD8, and ORD8 (H514A, H515A) mutant. *Error bars* indicate s.d.; $n = 3$. **b** Schematic of the assay employed to examine lipid transport by ORPs. See text for details. **c**, **d** Brain PtdIns(4)$P$ **c** and PtdIns(4,5)$P_2$ **d** transport assay. Donor liposomes ($L_A$) were incubated with NBD–PH$_{FAPP}$ followed by addition of acceptor liposomes ($L_B$) doped or not with PtdSer. After 3 min, the protein was injected. The *broken line* signifies NBD–PH$_{FAPP}$ signal upon complete PtdIns(4)$P$ and PtdIns(4,5)$P_2$ equilibration between liposomes. **e** Plot of initial brain PtdIns(4)$P$ and PtdIns(4,5)$P_2$ transport rates by Osh6p and ORP8 ORD. *Error bars* indicate s.d.; $n = 3$. **f** PtdIns(4,5)$P_2$ detected by PH-PLC–GFP in HeLa cells overexpressing ORP5A or ORP5A∆PH. **g** Quantitation of intensity in **f**, including the ratio of GFP fluorescence of the PM vs. cytosol, and the ratio of GFP signals detected by the TIRF microscopy vs. total epifluorescence (mean + s.d.; ***$P < 0.001$; *$P < 0.05$, $t$ test, $n = 8$ – 15 cells). **h** Western blot confirming the efficiency of ORP5 and ORP8 double knock-down in HeLa cells transfected with both GFP-ORP5A and mCherry-ORP8L. **i** Confocal microscopy images showing PtdIns(4,5)$P_2$ distribution as detected by PH-PLC–GFP in HeLa cells deficient in both ORP5 and ORP8. **j** Quantitation of intensity in **i**, including the ratio of GFP fluorescence of the PM vs. cytosol, and the ratio of GFP signals detected by the TIRF microscopy vs. total epifluorescence (mean + s.d.; ***$P < 0.001$; **$P < 0.01$, $t$ test, $n = 8$ – 15 cells). *Bar* = 10 μm

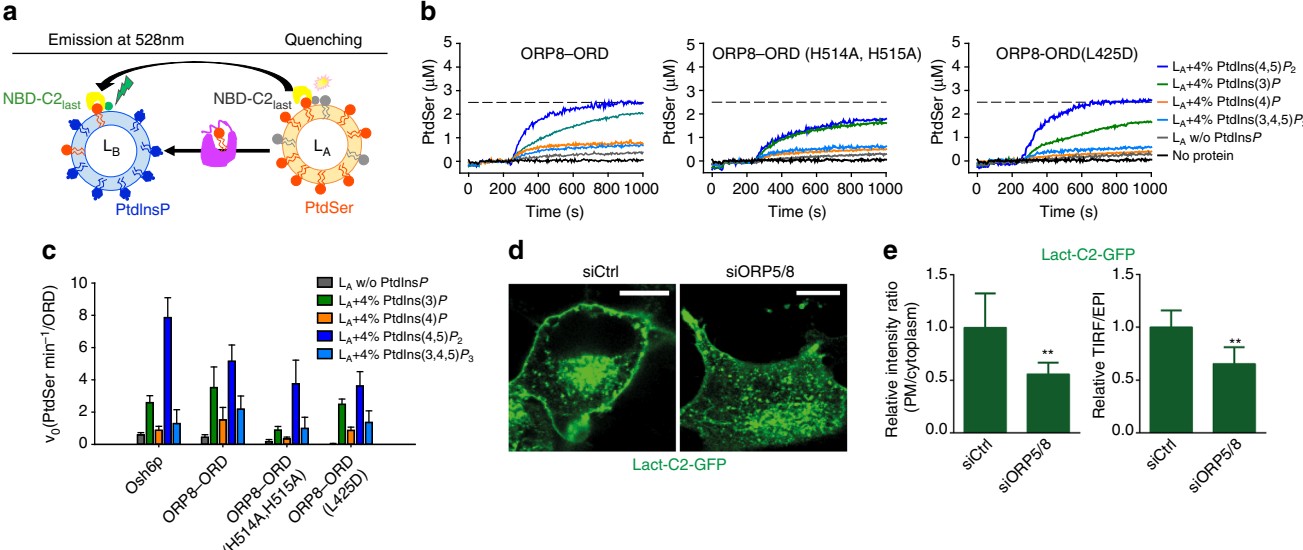

**Fig. 6** A PtdInsPs gradient is required for PtdSer transport by ORD8. **a** Schematic of the assay employed to study PtdSer transport by ORD8 under a PtdInsP gradient. **b** PtdSer transport assay. Donor liposomes ($L_A$) were incubated with NBD-C2$_{Lact}$ followed by addition of acceptor liposomes ($L_B$) doped with or without PtdInsPs. After 3 min, the transfer protein was injected. The *broken line* signifies NBD-C2$_{Lact}$ signal upon complete PtdSer equilibration between liposomes. PtdIns(3)$P$ and PtdIns(3,4,5)$P_3$ are 18:1/18:1. PtdIns(4)$P$ and PtdIns(4,5)$P_2$ are from brain. **c** Plot of initial PtdSer transport rates demonstrates ORP8 transports PtdSer more efficiently under a PtdIns(4,5)$P_2$ gradient. *Error bars* indicate s.d.; $n = 3$. **d** Distribution of PtdSer as detected by Lact-C2–GFP in HeLa cells deficient in both ORP5 and ORP8. **e** The ratio of GFP signals detected by the TIRF microscopy vs. total epifluorescence (mean + s.d.; **$P < 0.01$, $t$ test, $n = 6$ cells). *Bar* = 10 μm

remains to be firmly determined. To date, no structural information is available for the ORD of any of the mammalian ORPs, which may operate in a more complex way than their yeast counterparts. Clearly, much work is required to understand the targeting and cargo transport of ORP5/8. In summary, results presented here identify PtdIns(4,5)$P_2$ as a critical molecule for the ER–PM targeting of ORP5/8, and show, for the first time, that PtdInsPs other than PtdIns(4)$P$ may also serve as co-exchangers for the transfer of cargo lipids by ORPs.

## Methods

**Materials**. Dulbecco's Modified Eagle's Medium (DMEM), penicillin-streptomycin, and Dulbecco's phosphate-buffered saline (PBS) were obtained from Life Technologies Australia (Mulgrave, VIC, Australia). Fetal bovine serum (FBS) was obtained from Bovogen Biologicals (VIC, Australia). Rapamycin and protease inhibitor cocktail were obtained from Sigma-Aldrich (St. Louis, MO). All oligonucleotides were obtained from IDT (Integrated DNA Technologies) with standard desalting.

**Antibodies**. Antibodies used were goat polyclonal to ORP5 (catalog no. 59016, 1:250), mouse monoclonal to RFP (catalog no. 65856, 1:1000) purchased from Abcam, mouse monoclonal to GFP (catalog no. sc9996, 1:100) purchased from Santa Cruz Biotechnology, and mouse monoclonal to FLAG (catalog no. TA50011, 1:200) purchased from Origene. For immunostaining of plasma membrane PtdIns(4,5)$P_2$ and PtdIns(4)$P$, we obtained anti-PtdIns(4)$P$ IgM (catalog no. Z-P004, 1:62.5) and anti-PtdIns(4,5)$P_2$ IgM (catalog no. ZP045, 1:400) from Echelon Biosciences. For immunoblotting, we obtained horseradish peroxidise-conjugated secondary antibodies (catalog no. 715-035-150, 705-065-147, 1:5000) from Jackson ImmunoResearch. For immunostaining, we obtained Alexa Fluor secondary antibodies (catalog no. R57115 and A-21042, 1:500) from ThermoFisher Scientific.

**Lipids**. DOPC (1,2-dioleoyl-sn-glycero-3-phosphocholine) (catalog no. 850375P), DOPS (PtdSer for 1,2-dioleoyl-snglycero-3-phosphoserine) (catalog no. 850150P), brain PtdIns(4,5)$P_2$ (L-α-phosphatidylinositol-4,5-bisphosphate) (catalog no. 840046P), brain PtdIns(4)$P$ (L-α-phosphatidylinositol-4-phosphate) (catalog no. 840045P) liver PtdIns (L-α-PI) (catalog no. 840042P), 18:1/18:1 PtdIns(3)$P$ (1,2-dioleoyl-sn-glycero-3-phospho-(1′-myo-inositol-3′-phosphate)) (catalog no. 850150P), 18:1/18:1 PtdIns(3,4,5)$P_3$ (1,2-dioleoyl-sn-glycero-3-phospho-(1′-myo-inositol-3′,4′,5′-trisphosphate)) (catalog no. 850156P), Rhod-PE (1,2-dipalmitoyl-sn-glycero-3-phosphoethanolamine-N-(lissamine rhodamine B sulfonyl)) (catalog no. 810158P) were purchased from Avanti Polar Lipids. 16:0/16:0-PtdIns(4)$P$ (1,2-dipalmitoyl-snglycero-3-phosphoinositol-4-phosphate) (catalog no. P-4016),

16:0/16:0 PtdIns(4,5)$P_2$ (1,2-dipalmitoyl-sn-glycero-3-phospho-(1′-myo-inositol)-4,5 bisphosphate) (catalog no. P-4516) and the water-soluble diC8 phosphoinositides were purchased from Echelon Biosciences (USA). Inositol hexaphosphate (IP6) was purchased from Calbiochem (catalog no. 407125).

**Cell culture and transfection**. HeLa cells were obtained originally from ATCC. Monolayers of cells were maintained in DMEM supplemented with 10% FBS, 100 units/ml penicillin, and 100 μg/ml streptomycin sulfate in 5% $CO_2$ at 37 °C. DNA transfection was performed using Lipofectamine™ LTX and Plus Reagent (Life Technologies) according to the manufacturer's instruction. siRNA transfection was carried out in cells grown in full serum medium according to standard methods using Lipofectamine™ RNAiMAX transfection reagent (Life Technologies).

**Molecular biology and cloning**. For bacterial expression, cDNAs encoding human OPR5A (364–746), ORP8L (400–778) were cloned into a modified form of pET15b vector called pHUE for expression with an N-terminal HIS6 tag followed by an ubiquitin sequence. The tags were cleaved using a deubiquitinase enzyme (DUB), which specifically cleaves at the construct boundary without leaving any additional amino acids at the N terminus. Synthetic genes encoding ORP5A PH (127–243) and ORP8L PH (149–265) domain with an N-terminal MGSSGSSG and a C-terminal RSGPSSGLEEF linker were synthesized by Genscript and cloned into the pGEX-4T-2 plasmid for expression with an N-terminal GST-tag and thrombin cleavage site. All mutant constructs were generated using the QuikChange® Lightning Site-directed Mutagenesis Kit (Stratagene).

Osh6p was cloned into pCool vector with an N-terminal GST tag, ORP8 (331–835) was cloned into pET21b vector with a C-terminal HIS6 tag. The PH domain of the phosphatidylinositol-4-phosphate adapter protein-1 (FAPP, residue 1–100) was cloned into pCool vector. The PHFAPP was mutated to replace solvent-accessible cysteine by serine (mutation C37S and C94S) and to introduce a cysteine into a membrane-inserting wedge of the domain (T13C). The C2 domain of lactadherin (synthesized in SciLight Biotechnology, residue 270–427) was cloned into pCool vector with an N-terminal HIS6 tag, the solvent-accessible cysteine (C270, C427) was then mutated to alanine and, a cysteine was introduced into a region near the putative PtdSer-binding site (H352).

For mammalian expression, the constructs encoding GFP–ORP5, mCherry–ORP5, and mCherry–ORP8 encode full-length ORP5A or ORP8L with GFP or mCherry tagged to the N termini[17]. While the ORP5 and ORP8 point mutants were generated by site-directed mutagenesis, GFP–ORP5AΔPH, ORP5B, ORP8S, and N-terminally truncated ORP5A construct was made by deletion mutagenesis. MAPPER plasmid was kindly provided by Dr Jen Liou (UT Southwestern Medical Center)[18]. PJ (#37999), PJ-Sac (#38000), PJ-INPP5E (#38001), PJ-DEAD (#38002), LYN11-FRB–mCherry (#38004), and GFP–P4M-SidM (#51469) were obtained from Addgene. FLAG-PIP5K1b and GFP–PIP5K1b

were purchased from Genscript (catalog no. SC1691). PH-PLCσ1–GFP and PH-PLCσ1-RFP were gifts from Tamas Balla.

**Recombinant protein expression and purification.** The plasmids encoding ORP5A (127–243), ORP5A (127–243) (R136Q), ORP5A (127–243) (R179Q), ORP8L (149–265), ORP8L (149–265) (R158Q), ORP8L (149–265) (R201Q), ORP5A (364–746), ORP5A (364–746) (H478A, H479A), ORP8L (400–778), ORP8L (400–778) (H514A, H515A), ORP8L (331–835) (H514A, H515A), Osh6p, PH$_{FAPP}$, and C2$_{Lact}$ were transformed into BL21 (DE3)/Gold *Escherichia coli* cells. Cells were grown in 6-12L Luria Bertani broth at 37 °C until OD$_{600}$ reached 0.6. The temperature of the culture was brought down to 18 °C and the protein expression was initiated with 0.5 mM IPTG and allowed to grow at 18 °C overnight. The cells were harvested by centrifugation (5000 × *g*, 20 min, 4 °C). The cell pellet was resuspended in lysis buffer (20 mM Tris (pH 8.0), 500 mM NaCl, 100 units DNaseI, 1 tablet/50 ml EDTA free protease inhibitor (Roche), and 1 mM β-mercaptoethanol) for GST-tagged proteins and the lysis buffer was supplemented with 20 mM imidazole (pH 7.0) for HIS-tagged proteins. While, Ni–NTA resin (Qiagen) was used to purify HIS6-tagged proteins, glutathione sepharose beads (GE Healthcare) were utilized for purification of GST-fused proteins. The affinity tag was removed by adding 1 mg/ml TEV, thrombin, pre-scission or DUB wherever applicable. The cleaved proteins were eluted in 20 mM Tris, (pH 8.0), 200 mM NaCl, 20 mM imidazole (pH 7.0), and 1 mM mercaptoethanol. GST-cleaved proteins were eluted in 20 mM Tris, (pH 8.0), 200 mM NaCl, and 1 mM β-mercaptoethanol. The eluted proteins were further purified using gel filtration chromatography on a Superdex-200 column. For ITC experiments, proteins were gel filtered in 100 mM HEPES (pH 7.2) and 200 mM NaCl.

For crystallization, the cleaved ORP5A and ORP8L PH domain was purified on a Superdex-200 gel filtration column in 10 mM Tris (pH 8.0), 150 mM NaCl, and 1 mM DTT. The protein fractions were collected and concentrated to 15 mg/ml using a centricon.

For NBD labeling of PH$_{FAPP}$ and C2$_{Lact}$, after removing DTT, the proteins were mixed with a 10-fold molar excess of N,N′-dimethyl-N-(iodoacetyl)-N′-(7-nitrobenz-2-oxa-1,3-diazol-4-yl) ethylenediamine (IANBD-amide, Molecular Probes). The reaction was carried out overnight at 4 °C, and stopped by adding a 10-fold molar excess of L-cysteine. The free probe was removed by gel filtration and the labeled protein was analyzed by SDS-PAGE and UV-visible spectroscopy. The labeling yield (~100%) was estimated from the ratio of the optical density (OD) of tyrosine and tryptophan at 280 nm ($\varepsilon$ = 29,450 M/cm for PH$_{FAPP}$, $\varepsilon$ = 45,045 M/cm for C2$_{Lact}$), and NBD at 495 nm ($\varepsilon$ = 25,000 M/cm).

**Liposomes preparation.** Lipids from stock solutions were mixed at the desired molar ratio; the solvent was evaporated using nitrogen gas yielding a thin lipid film on the sides of a round bottom flask. The lipid film was thoroughly dried on a vacuum desiccator overnight. The films were hydrated in 50 mM HEPES pH 7.2, 120 mM potassium acetate, 1 mM MgCl$_2$ (HKM buffer) to obtain a suspension of multilamellar liposomes. The multilamellar liposome suspension was subjected to 12 freeze-thaw cycles using liquid nitrogen followed by extrusion through polycarbonate filters of 0.2 μm pore size using a mini-extruder (Avanti Polar Lipids). Liposomes were stored at 4 °C in the dark and were used within 2 days.

**Isothermal titration calorimetry.** The phospholipid-binding specificity and affinity of ORP5A PH, ORP5A ORD, ORP8 PH, ORP8 ORD, and various mutants were determined using a Microcal iTC200 instrument. Experiments were performed in the same buffer used for gel filtration buffer (diluted to 50 mM Tris (pH 8.0), 100 mM NaCl). The lipids at 1 mM or 0.5 mM were titrated into 0.025 mM proteins in 13 × 3.1 μl aliquots at 25 °C. Data were processed using ORIGIN to extract the thermodynamic parameters $\Delta H$, $K_a$ (1/$K_d$) and the stoichiometry n. $\Delta G$ and $\Delta S$ were derived from the relations $\Delta G = -RT\ln K_a$ and $\Delta G = \Delta H - T\Delta S$.

**Extraction assay.** The sample (600 μl) containing DOPC liposomes (150 μM total lipids), doped with 4% phosphoinositides, PtdSer (3 μM accessible) were mixed with NBD–PH$_{FAPP}$ or NBD-C2$_{Lact}$ (250 nM) at 25 °C in a small quartz cuvette. The NBD spectrum was recorded from 505 to 650 nm upon excitation at 490 nm before and 5 min after the injection of 3 μM proteins. The intensity at 528 nm measured before and after the addition of protein corresponds to $F_{max}$ and $F$. A control signal ($F_o$) was measured with the NBD–PH$_{FAPP}$ or NBD-C2$_{Lact}$ (250 nM) in buffer or in the presence of liposome with DOPC alone. The contribution of buffer or liposome alone was subtracted from the NBD signal. The percentage of extraction is calculated by 100×(1−(($F-F_0$)/($F_{max}-F_0$))).

**PtdSer transport assay.** A suspension (570 μl) of L$_A$ liposome (200 μM total lipids) containing 2% Rhod-PE and 5% PS was incubated with 250 nM NBD-C2$_{Lact}$ at 25 °C in HKM buffer under constant stirring. The concentration of accessible PtdSer (in the outer leaflet) is 5 μM. After 1 min, 30 μl of L$_B$ liposome (200 μM total lipids, final concentration) containing DOPC alone or with 4% PtdIns$P$s were injected. After additional 3 min, protein (200 nM) was injected. PtdSer transport was followed by measuring the NBD signal at 528 nm upon excitation at 460 nm. The NBD signal mirrors the distribution of NBD-C2$_{Lact}$ between L$_A$ and L$_B$

liposome. The amount of PtdSer transported by the protein is determined by normalizing the NBD signal. To that end, the NBD signal ($F_{eq}$) was measured for a condition where PtdSer is fully equilibrated between liposomes. NBD-C2$_{Lact}$ (250 nM) was mixed with L$_B$ and L$_A$ liposome (200 μM total lipid each) with a lipid composition similar to that of the liposomes used in the transport assay, except that each contains initially 2.5% PtdSer. The fraction of PtdSer on the L$_B$ liposome, PtdSer$_A$/PtdSer$_T$, is directly equal to the fraction of C2$_{Lact}$ on L$_B$ liposome and correspond to $F_{Norm} = 0.5 \times (F-F_0)/(F_{eq}-F_0)$ with $F_0$ corresponding to the NBD signal prior to the addition of protein. The amount of PtdSer transferred from L$_A$ to L$_B$ liposome corresponds to 5 × $F_{Norm}$[14].

**PtdIns $P$ transport assay.** For the PtdIns$P$ transport assay, a suspension (570 μl) of L$_A$ liposome (200 μM total lipids) containing 2% Rhod-PE and 4% PtdIns$P$ was incubated with 250 nM NBD–PH$_{FAPP}$ in HKM buffer under constant stirring. The concentration of accessible PtdIns$P$ (in the outer leaflet) is 4 μM. After 1 min, 30 μl of L$_B$ liposome (200 μM total lipids, final concentration) were injected. After additional 3 min, protein (200 nM) was injected. The NBD signal is measured with the same set-up as for PS transport assay; the NBD signal mirrors the redistribution of NBD–PH$_{FAPP}$ between L$_A$ and L$_B$ liposomes and was normalized to determinate the amount of PtdIns$P$ transported by the protein. NBD–PH$_{FAPP}$ (250 nM) was mixed with L$_A$ and L$_B$ liposome (200 μM total lipid each) that contains initially 2% PtdIns$P$. The fraction of PtdIns$P$ on the surface of L$_D$ liposome, PtdIns$P$ $_B$/PtdIns$P$ $_T$, is directly equal to the fraction of PH$_{FAPP}$ on L$_D$ liposome and correspond to $F_{Norm} = 0.5 \times (F-F_0)/(F_{eq}-F_0)$ with $F_0$ corresponding to the NBD signal prior to the addition of the protein. The amount of PtdIns$P$ (in μM) transferred from L$_A$ to L$_B$ liposomes corresponds to 4 × $F_{Norm}$.

**ORP8 PH domain crystal structure determination.** The protein was concentrated to 15 mg/ml and eight commercially available 96-well crystallization screens were set up using a Mosquito robot at 20 °C in the presence of 10 mM DTT. Plate crystals were obtained overnight in the screen containing 0.1 M HEPES (pH 7.5), 0.2 M ammonium formate, 0.2 M NSDB-195, and 27% PEG3350 by the sitting drop vapor diffusion method in a 24-well plate by mixing 1 μl of protein solution and 1 μl of mother liquor. Co-crystallization experiments of ORP8 PH domain-IP6 were also conducted. Briefly, 5 mM IP6 was mixed with the protein to a final concentration of 15 mg/ml and incubated at room temperature for 2 h. The mixture was then ultracentrifuged to get rid of any particulate matter. Sparse matrix screens were set up as for apo ORP8 PH domain and crystals grew after 2 days in a condition containing 0.1 M Bis-Tris propane, 0.2 M potassium thiocyanate, and 20% PEG3350.

The crystals were transferred to a cryo solution containing 20% glycerol in mother liquor, and cooled to 100 K under the cryostream. Data were collected at the Australian Synchrotron MX2 beamline. Data were integrated and scaled with iMOSFLM[40] and SCALA[41]. The ORP8 PH domain structure was solved by molecular replacement (MR) using PHASER[42], using the NMR ensemble of ORP8 PH domain as an input model (PDB ID 1V88). The MR solution was built using autobuild and the resulting model was rebuilt with COOT[43] followed by repeated refinement runs and model building with PHENIX[44] and COOT[43]. The PHENIX refinement protocol was comprised of isotropic refinement in combination with translation/liberation/screw (TLS) groups as well as individual and grouped B-factor refinement. The TLS groups were computed using the TLS server in the PHENIX suite. The final model contains one molecule in the asymmetric unit, a bound formate, NSDB-195 molecule, and 23 water molecules.

The diffraction data collected from ORP8 PH-IP6 co-crystallization crystals was processed and solved in the same way as ORP8 PH apo. ORP8 PH domain crystal structure (PDB ID 5U77) was used as the input model for MR. The final model contains four molecules in the asymmetric unit and 192 waters. Surprisingly, no density for the IP6 molecule was observed.

**Immunoblot analysis.** Samples were mixed with 2 × laemmli buffer, boiled for 5 min at 95 °C or incubated for 10 min at 70 °C, and then subjected to 7.5 or 10% SDS-PAGE. After electrophoresis, the proteins were transferred to Hybond-C nitrocellulose filters (GE Healthcare). Incubations with primary antibodies were performed at 4 °C overnight. Secondary antibodies were peroxidase-conjugated AffiniPure donkey anti-rabbit or donkey anti-mouse IgG (H+L; Jackson ImmunoResearch Laboratories) used at a 1:5000 dilution. The bound antibodies were detected by ECL western blotting detection reagent (GE Healthcare or Merck Millipore) and visualized with Molecular Imager® ChemiDocTM XRS + (Bio-Rad Laboratories) (Supplementary Fig. 11).

**Confocal microscopy.** Cells grown on coverslips were fixed with 4% paraformaldehyde for 15 min at room temperature. For immunostaining of plasma membrane PtdIns(4,5)$P_2$ and PtdIns(4)$P$, cells were fixed with 4% paraformaldehyde and 0.2% glutaraldehyde for 15 min at room temperature. Cells were washed three times with PBS containing 50 mM NH$_4$Cl. All subsequent steps were carried out on ice. Cells were blocked and permeabilized for 45 min in PBS containing 5% normal goat serum (NGS), 50 mM NH$_4$Cl, and 0.5% saponin. Primary antibodies were diluted in PBS containing 5% NGS and 0.1% saponin and applied to cells for 1 h. After three washes with PBS, cells were incubated with secondary antibody in

PBS containing 5% NGS and 0.1% saponin for 45 min. Next, cells were washed with PBS for four times and post fixed in 2% paraformaldehyde in PBS for 10 min on ice and 5 min at room temperature, followed by three washes with PBS containing 50 mM $NH_4Cl$. Cells were mounted in ProLong® Gold antifade reagent (LifeTechnology). Confocal images were acquired on an Olympus FV1200 laser-scanning microscope. Total internal reflection fluorescence (TIRF) and epi-fluorescence microscopy were carried out using a Zeiss Elyra microscope. The manufacturer's software and FIJI software were used for data acquisition and analysis.

**Statistical analysis**. Statistical analysis between groups was performed using Prism 6 for Windows Ver. 6.03 (GraphPad Software, San Diego, CA, USA) with Student's unpaired $t$ tests or one-way ANOVA. Data are expressed as mean + s.d. unless otherwise stated.

**Data availability**. Coordinates and structure factors for the ORP8 PH domain have been deposited at the Protein Data Bank (PDB) with accession codes 5U77 (ORP8 PH apo) and 5U78 (ORP8 PH co-crystallized with IP6). All the relevant raw data related to this study are available from the corresponding authors on request.

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

## Acknowledgements

We thank Brett Collins for his help in structure solving and for critically reading the manuscript. We also thank Viola Oorschot for assistance in electron microscopy

analyses. This work was supported by funds from the Australian Research Council (DP130100457), and National Health and Medical Research Council (NHMRC) of Australia (1041301) to H.Y. H.Y. is supported by an NHMRC Senior Research Fellowship. J.-W.W. is supported by Grant 2016YFA0502004 from the Ministry of Science and Technology of the People's Republic of China. R.G. is supported by an NHMRC-ARC Dementia Research Development Fellowship. The authors acknowledge the use of the Australian Microscopy & Microanalysis Research Facility at the Center for Microscopy and Microanalysis at The University of Queensland. R.G.P. was supported by grants and a fellowship from the National Health and Medical Research Council of Australia (Grant numbers APP1037320, APP1058565, and APP569542 to R.G.P.) and by the ARC Centre of Excellence in Convergent Bio-Nano Science and Technology.

## Author contributions

The X-ray crystallography, bioinformatics, structural modeling, and ITC studies were carried out by R.G. X.D. performed the cell biology experiments. Lipid extraction and transport experiments were designed and conducted by H.W., J.D. and J.-W.W. R.G.P. and C.F. performed the EM analyses. A.J.B., J.-W.W. and H.Y. conceived the project, and H.Y. coordinated the project and wrote the manuscript together with R.G. and all other authors.

## Additional information

**Competing interests:** The authors declare no competing financial interests.

