## [Peer Review File · Nature Communications]

Reviewers' Comments:

Reviewer #1:

Remarks to the Author:

In my previous review, I mentioned that some points that I raised could not be addressed because of some technical limitations (e.g. the availability of PIPs with defined acyl chains). Overall, and with this constraint, the authors performed a reasonable revision of a manuscript that was already of high quality. Therefore, I recommend publication.

Just one thing. I noticed in several circumstances a tendency to use strong words such as 'dramatic' 'strong' for changes that are significant but not huge (2 to 3 fold (see e.g. Figs 5G, 5J, 6D)). I don't think it is a good idea to over-emphasize the effects. The reader will make its own judgement when the effects are in this range.

Reviewer #2:

Remarks to the Author:

The authors have addressed all my concerns.

Reviewer #3:

Remarks to the Author:

The authors have addressed my concerns fully, and I consider this good and interesting and important work.

Some very minor points:

Although there are a few other typos in the ms, two that the copy editors might not catch are in the legend for Figure 4 and and Suppl Figure 8, p. 25 and 28 of ms. In both cases, "25 mM" should probably be "25 microM".

Also, to be consistent, it would be best to keep LA for donor liposomes and LB for acceptor liposomes (or vice versa) throughout the paper, rather than one way for some experiments and another for the others.

Point-by-point responses to reviewers' comments:

Reviewer #1 (Remarks to the Author)

In my previous review, I mentioned that some points that I raised could not be addressed because of some technical limitations (e.g. the availability of PIPs with defined acyl chains). Overall, and with this constraint, the authors performed a reasonable revision of a manuscript that was already of high quality. Therefore, I recommend publication.

Just one thing. I noticed in several circumstance a tendency to use strong words such as 'dramatic' 'strong' for changes that are significant but not huge (2 to 3 fold (see e.g. Figs 5G, 5J, 6D). I don't think it is a good idea to over emphasize the effects. The reader will make its own judgement when the effects are in this range.

Answer: We thank the referee for carefully reviewing our manuscript. We have now changed these wordings.

Reviewer #2 (Remarks to the Author)

The authors have addressed all my concerns.

Answer: We thank you for encouraging comments.

Reviewer #3 (Remarks to the Author)

The authors have addressed my concerns fully, and I consider this good and interesting and important work.

Some very minor points:

Although there are a few other typos in the ms, two that the copy editors might not catch are in the legend for Figure 4 and and Suppl Figure 8, p. 25 and 28 of ms. In both cases, "25 mM" should probably be "25 microM".

Also, to be consistent, it would be best to keep LA for donor liposomes and LB for acceptor liposomes (or vice versa) throughout the paper, rather than one way for some experiments and another for the others.

Answer: Thank you for your kind words and for highlighting the typo errors in the text. We have now rectified these errors. The naming of the liposomes has also been made consistent throughout the manuscript.